# The Role of Experience, Perceived Match Importance, and Anxiety on Cortisol Response in an Official Esports Competition

**DOI:** 10.3390/ijerph18062893

**Published:** 2021-03-12

**Authors:** Guillermo Mendoza, Vicente Javier Clemente-Suárez, José Ramón Alvero-Cruz, Iván Rivilla, Jerónimo García-Romero, Manuel Fernández-Navas, Margarita Carrillo de Albornoz-Gil, Manuel Jiménez

**Affiliations:** 1Departamento de Fisiología Humana, Histiología, Anatomía Patológica, y Educación Física y Deportiva, Universidad de Málaga, 29010 Málaga, Spain; gmendoza.t@gmail.com (G.M.); alvero@uma.es (J.R.A.-C.); jeronimo@uma.es (J.G.-R.); marcargil@uma.es (M.C.d.A.-G.); 2Faculty of Sports Science, Universidad Europea de Madrid, 28670 Madrid, Spain; vctxente@yahoo.es; 3Grupo de Investigación en Cultura, Educación y Sociedad, Universidad de la Costa, Barranquilla 080002, Colombia; 4Departamento de Didáctica de la Educación Física y Salud, Universidad Internacional de La Rioja, 26002 Logroño, Spain; ivan.rivilla@unir.net; 5Departamento de Didáctica y Organización Escolar, Universidad de Málaga, 29010 Málaga, Spain; mfernandez1@uma.es

**Keywords:** esport, stress, competition, cortisol, anxiety

## Abstract

The aim of the present study was to analyse the neuroendocrine stress response, psychological anxiety response, and perceived match importance (PMI) between expert and non-expert control gamers in an official competitive context. We analyzed, in 25 expert esports players and 20 control participants, modifications in their somatic anxiety, cognitive anxiety, self-confidence, PMI, and cortisol in a League of Legends competition. We found how expert esports players presented higher cortisol concentrations (Z = 155.5; *p* = 0.03; Cohen’s d = −0.66), cognitive anxiety (Z = 99.5; *p* = 0.001), and PMI (Z = 50.5; *p* < 0.001) before the competition than non-experts participants. We found a greater statistical weight in the cognitive variables than in the physiological ones. The results obtained suggest that real competitive context and player’s expertise were factors associated with an anticipatory stress response. The PMI proved to be a differentiating variable between both groups, highlighting the necessity to include subjective variables that contrast objective measurements.

## 1. Introduction

Electronic sports (esports) have increased in popularity, number of players, followers, and sponsors in recent years [1]. Although there is still no consensus on its exact definition, many authors agree on the competitive aspect that the term implies [2,3,4,5,6,7,8]. The competitive structure based on rankings and tournaments and governed by official institutions makes the original videogame into an esport [1,6]. The requirements of esports in the competitive context mean esports players need to develop skills like working memory, inhibitory control, and cognitive flexibility, among others [9]. Trying to understand the role of these skills in esports performance to develop specific training methods allows us to apply traditional sport theory to esports [10]. In this line, the different physiological responses of gamers depending on the type of game were highlighted, showing how first-person shooter games elicited a larger change than multiplayer online battle arena games [11].

League of Legends (LOL) is one of the most popular video games in the world, with more than 100 million players [12]. It is also one of the video games with more academic studies [13]. LOL is classified as a multiplayer online battle arena game (MOBA) that consists of forming teams of five players whose objective is to destroy the enemy “nexus” (main structure in their base). Each player must choose a character (i.e., champion) from among 152 possible characters with different characteristics and abilities, which provides a great variety of combinations, making LOL a strategically complex game. This complexity demands a great amount of time from players to prepare and train for competitions [4,14].

Sports competitions are a source of stress that provoke different neuroendocrine responses depending on the competitive moment: before, during, and after the competition [15,16,17,18,19]. An increase in cortisol (C) and sympathetic autonomic branch has been observed prior competitions, highlighting the anticipatory response prior to the competition due to the psychological anticipation influenced by Competitive State Anxiety Inventory-2 (CSAI-2) factors about subjective feelings of anxiety and confidence [20,21,22,23]. The anticipatory stress response can be observed from days before the stressor [24,25], but it is when the competitions approach that we find the biggest manifestations [26]. Cortisol is the primary stress hormone, and increases sugars (glucose) in the bloodstream, enhances the brain’s use of glucose and increases the availability of substances that repair tissues. Cortisol also curbs functions that would be nonessential or detrimental in a fight-or-flight situation.

The anticipatory stress response previously mentioned could be somatized as a subjective increase in anxiety. Prior research has found how demanding situations such as sport competitions, ultraendurance events, exams, academics exposition, parachuting, or events before military missions can be—contexts wherein the perception of uncontrollability and uncertainty trigger not only the hormonal stress response, but also the cognitive resources that prepare us to face the possible threat or challenge. In this line, the anxiogenic response was normally measured using questionnaires or validates scales like the Competitive State Anxiety Inventory-2 (CSAI-2) or the State-Trait Anxiety Inventory (STAI).

Previous studies have already addressed cortisol response analysis in esports, obtaining different results [3,27,28,29,30,31,32,33,34,35]. Some studies have observed moderate decreases in cortisol during unofficial competitive games [13,28], where winning or losing had no direct influence on their social status concerning other participants. On the contrary, other studies have observed increases in cortisol, especially in higher-ranked players [29,30,31,32,33,34,35,36], also observing increases in heart rate both in expert players and between competition days and rest days [3,29]. This variability in results could be explained by the different methodologies used [26]. These studies analysed recreational players and simulated competitions with little stimulating prizes [37]. In addition, the players’ expertise level was not considered as a factor that can modulate the performance and stress responses of players, especially as previous studies found that expert gamers present different brain plasticity [5] and different responses during “low time pressure events” [38], “multiple object tracking” [39], and intelligence tests [40] than non-expert gamers.

For a better understanding of neuroendocrine responses in competition, previous studies [15] have suggested complementing observational methods (victory or defeat) with the analysis of the perceived importance, control, and subjective experience of participants. In this line, we present the current research with the objective to analyse the neuroendocrine stress response, psychological anxiety response, and perceived match importance (PMI) between expert and non-expert control gamers in an official competitive context.

## 2. Materials and Methods

### 2.1. Participants

We studied 45 volunteer male gamers from 18 to 27 years old. The sample was divided into two groups. A control group (*n* = 20) had participants who had never before played LOL or any other MOBA, first person shooters, or real-time strategy games; although some of them played other types of videogames before such as casual mobile games, none of them consider themselves gamers nor engaged on any esports activities in the past. The other group consisted of expert gamers (*n* = 25) who competed in official esports tournaments. The minimum level of expert players at the time of the study was the Platinum ranking (internal game ranking), which places them in the 85th percentile of all users [41]. Because all experts were competitive players, participating in tournaments with established teams, they play and train between 15 and 40 h weekly (mean = 25 ± 3).

### 2.2. Procedure

To reach the study aim, we gathered data during two face-to-face official competitions with incentives and prizes that could encourage motivation to win, such as gaming gear (keyboard, mouse, and backpack) and up to 1000€ in prize money for the champion. In the first competition, we analyzed 10 expert gamers and, in the second one, another 15 expert gamers. For control participants, they were equally divided in four teams and played against each other. As they had never played before, the basics of the game were explained to them and they were given a few minutes to familiarize themselves with the controls. Expert gamers had been playing MOBA games for more than 6 years, and had more than 5 years of experience in professional or semi-professional teams in official LOL national and international leagues (i.e., Superliga Orange, Spain; Iberian Cup, Spain; and European Championship). In order to have the closest conditions to a real competition, all the games were placed in the same room in which the competitions were played by the experts participants, and winners were decided in the same way as those official tournaments, by winning two of three games (BO3).

All participants played a total of two games, both against their respective rival team. Because all winners won their first two games, none of the matches needed a third tiebreaker game. After the first game was over, the second started immediately. Each game lasted between 30 and 60 min. On competition day, 30 min before the games, all participants completed the Competitive State Anxiety Inventory-2 (CSAI-2) [42], which is divided into three subscales: somatic anxiety, cognitive anxiety, and self-confidence. To be sure that the challenge of winning the match was stimulating enough for the participants, a question was posed about PMI with a Likert scale response between 1 (low) and 10 (high). Participants were instructed not to engage in vigorous physical exercise, consume caffeinated beverages, or eat food for at least 2 h prior to the games. Saliva samples were taken 10 min before and 10 min after the games were concluded. To analyze the hormonal responses, Super.Sal^®^ devices (Oasis Diagnostics, Washington, USA) were used, which consist of an aseptic cotton swab that is placed in the mouth for 1 min and collects a basic sample of saliva with a visual indicator, allowing confirmation that a homogeneous sample of 1–2 mL was obtained. Cortisol concentrations were determined before and after each game. All participants’ saliva samples were taken between 10:00 and 14:00, immediately frozen at −40 °C, and stored in the laboratory of the University of Malaga. Samples were centrifuged for 15 min at 3000 rpm and immunoassayed using the Grifols Triturus^®^ (Somagen Diagnostics Inc., Edmonton, AL, Canada) equipment and competitive enzyme immunoassay kits (Diametra, Milan, Italy) with inter-assay coefficients of variation of 9.6%, sensitivity 0.5 ng/mL, and detection limits of 100 ng/mL for cortisol. Samples were immunoassayed twice.

### 2.3. Statistical Analysis

Shapiro–Wilk test was applied to all variables. Cortisol, cognitive anxiety, and PMI were not normally distributed; therefore, nonparametric methods were used for further analysis. Mann–Whitney U test was used to compare data from both groups and Spearman’s rank correlation coefficient was applied to test the relation between variables. Effect sizes were also calculated using Cohen’s *d*, and all statistical analyses were considered significant with *p* < 0.05. All statistical analyses were performed with SPSS^©^ 20.0 statistical package (IBM Co, Armonk, NY, USA).

## 3. Results

### Cortisol

Figure 1 shows the anticipatory stress response of expert and control players. The Mann–Whitney test shows that cortisol levels before playing were higher in the experts (Z = 155.5; *p* = 0.03; Cohen’s d = −0.66), while after playing, there were no differences. The analysis between winners and losers showed no significant results on cortisol.

Pre-game cortisol levels were positively correlated with self-confidence (r = 0.470, *p* = 0.001) and with PMI (r = 0.411, *p* = 0.005), and negatively with somatic anxiety. At the end of the games, only self-confidence was correlated with post-game cortisol (Table 1).

Experts presented a higher score in cognitive anxiety (Z = 99.5; *p* = 0.001), evaluating their games with higher PMI than the control group (Z = 50.5; *p* < 0.001) (Table 2). As with cortisol, the analysis between winners and losers did not yield any significant results on these variables.

Correlational analysis indicated a positive correlation between cognitive anxiety and PMI (r = 0.557; *p* < 0.001), as well as a negative correlation between somatic anxiety and self-confidence (r = −0.766; *p* < 0.001).

## 4. Discussion

The aim of this study was to analyze the neuroendocrine stress response, psychological anxiety response, and PMI between expert and non-expert control gamers in an official competitive context. The expert group only presented significantly higher cortisol levels in pre-competition situations. Cognitive anxiety and PMI were higher in the expert group, but not somatic anxiety and self-confidence. This comparative design between experts and a control group used a real competitive context, no fit with single group, and competitive laboratory or friendly conditions [37]. We found that expert esports players presented higher level of precompetitive cortisol than controls. This result was in line with previous studies that found how participation in real esports competitions produced a large state of physiological arousal [36]. However, the effect size of pre-competition cortisol was small, so they should be taken with caution. This similar result may be due to the experimental design conducted, as the control group experiment took place in a context as close as possible to a real competition. The usual pre-competition “warm-up” performed by experts was also implemented in the control group, where they learn the controls and become familiar with the game. This is an important factor as warm-up time is a factor that could affect the cortisol levels in athletes [43]. In addition to the game mode to determine the winner (BO3), the environment and computers used were the same in both the expert and control groups. With this strategy, we could have minimized variability in both groups.

We did not find significant modifications of cortisol between the pre and post competition samples, as well as between winners and losers. These results were in line with previous authors who did not find significant modification of the cortisol hormone [28,29,30]. Nevertheless, other studies in sports competitions reported increases in cortisol when losing and decreases when winning, but modifications in cortisol levels after winning or losing in non-athletics competitions are still uncertain [22].

Expert esports players presented significantly higher cognitive anxiety and PMI than the control group before the game, with no significant differences in somatic anxiety and self-confidence. These results suggest that expert esports players have a higher psychological expectation placed on their own performance prior to the competition. This result reinforces previous postulations that evidenced the importance of psychological factors on having a better understanding of the physiological changes that esports players experience during a competition [1,22,37]. This anticipatory anxiety response was also evaluated in different sports competitions, prior to stressful context, such as military actions or even prior to academic evaluations or clinical stays [44,45]. We found how different stimuli elicit a response that prepares the organism both physically and psychologically for a stressor, with the stressor of the esports competition causing a response in line with that evaluated in traditional sports.

The lack of differences in self-confidence and somatic anxiety between both groups could be explained by the subjectivity in the perception of these parameters, causing a greater variability in the responses [16,17,46]. as similar levels of expertise among rivals were controlled, perceived expertise could no longer be a factor affecting the self-confidence of participants. In this line, the self-confidence of control participants might not have been affected during competition as they knew that they were going to play against players with a similar level. Other factors not measured in this study could have a greater influence on the self-confidence of participants, such as mistakes committed previously in other matches [18,46], and are advised to be included in future research. In this line, somatic anxiety is also affected by the individual subjectivity, as previous studies reported how perceived arousal levels are not related to the objective measured arousal [23].

Correlational analysis was consistent with the previous research, as pre-competition cortisol was correlated with self-confidence and PMI. Participants with a low anxiety trait showed higher levels of both cortisol and self-confidence in competitive settings [10]. The negative correlation of cortisol with somatic anxiety may be related to the high values obtained by control participants in this factor. Despite that control participants did not present cognitive anxiety and did not evaluate the competition as important, it was in fact their first time in an esports arena where official competitions take place. They also played for the first time with gaming computers and competitive equipment, following the same protocol as the experts, as well as tournament rules. All these factors may have caused some activation at peripheral level that was not reflected at the cognitive level.

Although both hypotheses were partially confirmed, the expert group clearly differed from the control group. Skilled players have been differentiated from recreational players in previous studies [2,3,5,39,40], highlighting the importance of analyzing competitive experienced players and not only recreational players in esports research. On the other hand, because we found pre-competitive differences, but not after the competition, it is interesting to propose that future studies report the competitive moment in which the players are tested, as it is possible to obtain different results depending on the proximity of the next competition. Finally, for a better understanding of anticipatory stress responses in esports players, different levels of experience and different competitive contexts should be included in future research.

The principal limitations of the present research were the low sample size used, limited due to the complications of recruiting specific player in this collective—a fact that also precluded a randomized participants selection system. The effect size of cortisol results was small, evidencing the necessity of further study, considering factors such as previous results or perceived capacity. Finally, the design did not contemplate a baseline data collection—a fact that could improve information about the modification obtained in competition. The control group participants played against others in the control group, and this situation could have affected the results.

## 5. Conclusions

Expert esports players presented higher cortisol concentrations, cognitive anxiety, and PMI before the competition than non-expert participants. We found a greater statistical weight in the cognitive than physiological variables. The results obtained suggest that real competitive context and players’ expertise were factors associated with an anticipatory stress response. The PMI proved to be a differentiating variable between both groups, highlighting the necessity to include subjective variables that contrast objective measurements.

## Figures and Tables

**Figure 1 ijerph-18-02893-f001:**
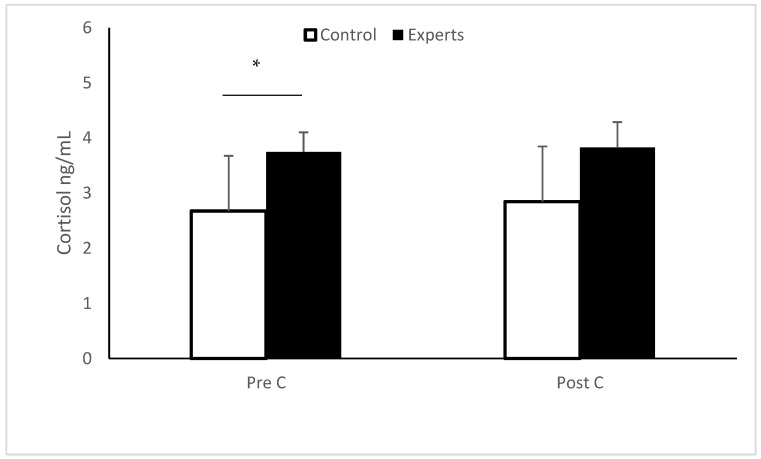
Mean ± SD for salivary cortisol levels between control and experts players. Pre C = pre-competitive salivary cortisol levels; Post C = post-competitive salivary cortisol levels. * *p* < 0.05.

**Table 1 ijerph-18-02893-t001:** Mean ± SD for salivary cortisol, anxiety, and perceived match importance differences between experts and control group (*n* = 45).

Dependent Variables	Experts (*n* = 25)	Control (*n* = 20)	*p*
Cortisol Pre-Game (ng/mL)	3.75 ± 1.76	2.68 ± 1.48	0.03
Cortisol Post-Game (ng/mL)	3.83 ± 2.30	2.85 ± 1.35	0.11
Cognitive Anxiety	22.44 ± 7.41	15.70 ± 3.99	0.001
Somatic Anxiety	26.56 ± 8.74	22.65 ± 9.43	0.166
Self Confidence	27.24 ± 8.77	24.40 ± 8.80	0.314
Perceived Importance (1–10)	6.04 ± 2.05	2.55 ± 1.70	<0.001

Anxiety, confidence, and perceived match importance.

**Table 2 ijerph-18-02893-t002:** Correlation matrix for cortisol, cognitive anxiety, somatic anxiety, self-confidence, and perceived game importance.

	1	2	3	4	5	6
1. Pre Cortisol	1					
2. Post Cortisol	0.56 **	1				
3. Cognitive Anxiety	0.15	−0.04	1			
4. Self-Confidence	0.47 **	0.45 **	0.11	1		
5. Somatic Anxiety	−0.31 *	−0.22	0.23	−0.77 **	1	
6. PMI (1–10)	0.41 **	0.26	0.56 **	0.19	0.05	1

Pre cortisol = pre-competitive salivary cortisol; post cortisol = post-competitive salivary cortisol; PMI = perceived match importance. * *p* < 0.05, ** *p* < 0.01.

## Data Availability

Data are not available to prevent professional players´confidenciality.

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
