# Peer review of "The Role of Experience, Perceived Match Importance, and Anxiety on Cortisol Response in an Official Esports Competition"

_ijerph, 2021, doi:10.3390/ijerph18062893_

Round 1
Reviewer 1 Report
Dear Authors,
You have written an interesting paper, however, some parts need to be addressed for greater clarity.
The introduction is too short and it does not address all of the necessary topics. Anxiety and how it is measured-tested in sport or esports in not well described. ADD
Cortisol role in human physiology is not stated. Add
You stated - ''Previous studies have already addressed cortisol response analysis in esports obtaining different results'' so what are these different results? report them or at least the range of measured values.
You also didn't mention that there are differences in game styles as FPS, RTS and MOBA on a physiological and psychological level. I recommend some current literature: https://doi.org/10.3389/fpsyg.2020.01030
The sentence in lines 63 to 65 is not finished - correct.
Sample:
Inclusion criteria for the control group need to be better described: What is their experience with gaming overall. Did they maybe have experience with FPS or RTS style of games? What about the experience in other MOBA games? report
The expert group need some additional data: what is their overall gaming experience, how long are they playing MOBA style games, how long are they playing LOL, how many hours per day are the playing-training LOL - ADD.
Procedure: Why did you choose for the control group to play between themself? When beginners play against beginners their anxiety levels and physiological responses are different in comparison to when playing against experts. This influences your results in a great matter. Elaborate and back up your decision.
Limitations of the study should be better addressed. Especially the sample selection and the testing procedure.
The paper needs additional work and I recommend major revision.
Kind regards
Author Response
Dear Reviewer # 1
Thank you very much for taking the time to review and provide feedback for the improvement of the manuscript. I have carefully read all your recommendations and I believe that all of them should be heeded for a better understanding of the research we have presented. I hope I have responded correctly to all your comments in the final document that I attach. Again, I would like to thank you for the elegance and respect of all your comments.

Reviewer 2 Report
The current paper needs extensive reworking on the grammar and formatting. The abstract needs much more information with result values being reported.
Lines 49-54 give more context and information on the importance here
line 80 how much money?
line 81 make sure you display numbers the same way each time
line 88 no games went to a third trial for the best of three?
line 100 approximately, any more information on running replicates and such?
figure 1, improve your formatting here
line 140 is this rho or r?
line 132 what are these z values? how are you referring to them in the table?
line 166 you need to report this data in the results
line 203 "recreative"? do you mean recreational?
the statement in the conclusion has the previous paragraph of the discussion disagreeing with it.
Author Response
Dear Reviewer # 2
We appreciate your review and the detailed comments. We have carefully read each one because we consider that comments can all bring a clear improvement to the manuscript. I beg you to observe in the revised manuscript how we have attended to each one of them because we are sure that it has improved substantially. Thanks for your time, kindness, and great advice.
Cordially

Round 2
Reviewer 1 Report
Dear Authors,
Thank you for addressing the raised issues and questions. In my opinion, your paper is now in a suitable form for publication.
Kind regards